# Feasibility Study of a Small-Scale Recirculating Aquaculture System for Sustainable (Peri-)Urban Farming in Sub-Saharan Africa: A Nigerian Perspective

**Emmanuel O. Benjamin** [1,*], **Oreoluwa Ola** [2] and **Gertrud R. Buchenrieder** [3]

1   Department of Agricultural Production and Resource Economics, Technical University of Munich (TUM), 85354 Freising, Germany
2   Department of Governance in International Agribusiness, Technical University of Munich (TUM), 85354 Freising, Germany
3   RISK Research Center, Universität der Bundeswehr München (UniBw M), 85577 Neubiberg, Germany
*   Correspondence: emmanuel.benjamin@tum.de

**Abstract:** The (peri-)urban population in developing countries, especially sub-Saharan Africa, is rapidly increasing. As towns and cities grow, so does the demand for fish protein. While flow-through aquaculture can provide fresh, healthy and nutritious fish protein, it is plagued by extensive land requirements as well as effluent discharge and is thus unsuitable for city regions. Alternatively, small-scale Recirculating Aquaculture Systems (RAS) could improve food and nutritional security and livelihoods as well as reduce environmental degradation in (peri-)urban areas despite land and water constraints. The question, however, remains—what are the key technical, business and managerial issues surrounding small-scale RAS in (peri-)urban farming? To answer this question, first, a systematic literature review on RAS in sub-Saharan Africa is conducted. Second, the RAS prototype of the Sustainable Aquaponics for Nutritional and Food Security in Urban Sub-Saharan Africa (SANFU) II project is assessed. This assessment is based on the mass balance and stock density, relevant for fish survival and/or availability as well as net cash flow analyses. The results suggest that small-scale RAS are technically and financially viable with efficient filtration and family labor having proper aquaculture monitoring and management skills. Furthermore, access to adequate equipment and inputs as well as electricity for the recirculating system are crucial. (Peri-)urban innovation actors will adopt RAS if operations are profitable.

**Keywords:** food security; (peri-)urban farming; fish protein; RAS; land; water; Nigeria; sub-Sahara Africa



## 1. Introduction

The increasing population and urbanization in sub-Saharan Africa is reaching unprecedented levels. In a number of instances, urbanization has passed the 50% threshold [1]. A number of factors ranging from high fertility rates to rural–urban migration drives population growth and urbanization [2,3]. This implies that there will be a substantial increase in the demand for animal-sourced foods (ASF), specifically fish protein, in cities and towns, which today's (rural) producers will be unable to meet [4]. Furthermore, prospective (rural) fish farmers, often young and agile, seek greener pastures in cities abandoning farming altogether. This exacerbates the fish supply deficit observed in sub-Saharan Africa, particularly in Nigeria, where the fish consumption rate is less than 10 kg compared to the world average of 19 kg per person per year [5]. This is more severe among women and children, resulting in malnutrition and one of the highest stunting rates among children under the age of five [6]. To compensate for this deficit, (peri-)urban vulnerable groups often turn to backyard gardens and other micro- and small-scale agribusinesses for subsistence as well as income generation [7]. Thus, one of the alternatives and viable options for ensuring food

and nutrition security (FNS) and alleviating poverty in (peri-)urban areas that has gained recognition over the years is (peri-)urban farming. The term (peri-)urban farming implies the cultivation of plants and the raising of animals for food in cities and towns (similar to the definition of urban agriculture by De Bon et al. [8]). Urban and peri-urban farming has to be embedded in urban planning and development as well as urban–rural interactions given its demand for scarce resources such as land as well as the corresponding water, energy, food and ecosystem resources, the so-called WEFE Nexus.

Conventional (peri-)urban farming related to (flow-through) aquaculture has always been relevant for animal protein in and around cities and towns in developing countries, specifically, Lagos, Nigeria [2]. However, flow-through aquaculture involves the consistent exchange of fish wastewater for maintaining water quality levels and fish health. This leads to a high discharge of effluents (effluent is any form of liquid waste that is discharged into a body of water) into the ground water as well as other water bodies, resulting in eutrophication (eutrophication is the prevalence of excessive nutrients in a body of water). The increasing rate of urbanization, particularly in sub-Saharan Africa, raises questions about the ability of flow-through aquaculture to be viable given land and water constraints as well as environmental concerns. Climate change introduces additional challenges to flow-through aquaculture as the competition for water and ecosystem services intensifies and non-circular food systems heighten the strained relationships between the WEFE Nexus. However, limited efforts in improving productivity and efficiency while reducing environmental pollution in conventional aquaculture in African cities have been made [9–13]. The ongoing COVID-19 pandemic, and more recently the Russian–Ukrainian conflict, have further aggravated the food and nutrition insecurity of (peri-)urban dwellers in sub-Saharan Africa [14]. Both external shocks disrupted regional and international supply and value chains and caused food and energy price inflation. This experience revealed the need to make local and regional food systems more crises-proof, which means increasing system resilience while conserving resources and the environment.

Thus, to improve (peri-)urban FNS and to reduce the pressure on the environment, Davies et al. [9] and FAO [15] call for the introduction of novel circular agri-food technologies and practices in the (peri-)urban food systems, which simultaneously generate income through profitable businesses. A number of studies [4,16–18] have also emphasized the need for innovative (peri-)urban farming technologies that require limited resources (i.e., land and water). Examples of such innovations include recirculating aquaculture systems (RAS). These technologies are suitable for African (peri-)urban settings because they do not require great access to land, water or wealth. Moreover, they can spur job creation, especially for women and young adults. According to Fornshell and Hinshaw [11] (p. 11), "recirculating aquaculture systems consist of a culture unit connected to a set of water treatment units that allows some of the water leaving the culture unit to be reconditioned and reused in the same culture unit. Recirculating aquaculture systems minimally require water treatment processes to remove solids, remove or transform nitrogenous wastes, and add oxygen to the water". However, the rather high up-front costs of RAS and the operating costs related to electricity, which is essential to maintain the water circulation and aeration, have limited its adoption among (peri-)urban farmers [11]. Furthermore, Ahmed and Turchini [19] and Bodiola et al. [20] argue that the rather complex fish production technology limits the adoption and implementation of RAS in developing countries, especially in those of sub-Saharan Africa. Bodiola et al. [20] also attribute the lack of skilled staff for water quality control and repair of mechanical faults to the slow adoption of RAS. Despite all its advantages with regard to improving FNS as well as the WEFE Nexus, the relatively high up-front costs and the related delay in the payback period may discourage poor and vulnerable (peri-)urban dwellers to adopt RAS. Nevertheless, Aich et al. [21] argue that RAS has the potential to produce 30–50 times more fish per unit area compared to conventional fish farming. However, the economic viability of relevant parameters such as optimal and maximum density, market prices, energy cost, etc., are often based on best guesses. This implies that data, which provide insights on the challenges and op-

portunities of (peri-)urban RAS adoption and implementation, are lacking. Thus, RAS has not witnessed broad adoption and implementation in (peri-)urban farming in developing countries, especially in sub-Saharan Africa [16–21].

This leads to the research question: what are the key opportunities and challenges of small-scale RAS implementation in (peri-)urban farming contexts from a technical, business and managerial perspective in sub-Saharan Africa?

To answer this research question, this study first conducts a systematic literature review of RAS, and secondly, revisits the design and technical details as well as costs and benefits of micro- and small-scale RAS in Africa. We explore the results for one fish production cycle of the Sustainable Aquaponics for Nutritional and Food Security in Urban Sub-Saharan Africa II (SANFU II) project from March to June 2022. The SANFU II RAS prototype is a simple micro- and small-scale RAS (600-L fish tank, sorting and sump tanks with 148 African Catfish—*Clarias Garipinus*) undergoing testing in Lagos, Nigeria. The design, capacity and cost related to setting up and managing the SANFU II RAS are presented.

In a nutshell, the results suggest that micro-and small-scale RAS can be stocked at a density that is higher than conservative recommendations given an efficient filtration system is in place. This relative high stocking density requires certain management skills and entails a higher risk of fish mortality. The monthly fixed and variable costs associated with running the RAS for a complete fish production cycle of four months were estimated at ₦36,733 (US$63) and ₦16,733 (US$29), respectively, and can be reduced if managed by skilled family members. Unemployment rates in developing countries in sub-Saharan Africa are rather high: the unemployment rate in Nigeria reached 33%, the youth unemployment rate, 42.5% (2021 and 2022, www.statista.com/statistics, accessed on 18 October 2022). This does not account for those people who might want to work but are not actively searching for employment. They are classified as discouraged job seekers or as hidden unemployed. Both forms of unemployment reflect an important loss of productive capacity, loss of national income, and issues of social exclusion [22]. Thus, smallholder farming in (peri-)urban areas can bring the unemployed family members back into the productive production process. While their labor may not be paid according to their marginal productivity if employed in the so-called first labor market, they are paid according to the average productivity of all family members as they raise the overall (subsistence and cash) family income.

The paper is structured as follows: Section 2 presents the materials and methods used. A systematic literature review is summarized in Section 2.1 and the technology and business management analysis of small-scale RAS in (peri-)urban farming is outlined in Section 2.2. Section 3 presents the results of the analysis. Section 4 provides insightful discussion of the results before concluding in Section 5.

## 2. Materials and Methods

### 2.1. Literature Review—RAS Performance, Opportunities and Challenges in Africa

Different forms of aquaculture have existed for centuries. Nevertheless, conventional flow-through aquaculture relies on abundant supplies of water and land and contributes to the generation of greenhouse gas emissions and loss of biodiversity. RAS expands the frontiers of current food production practices, especially with regard to reducing the pressure on the WEFE Nexus. Yet, while RAS has gained traction worldwide, its adoption and implementation rate is still rather modest on the African continent. Therefore, the key technical, business and managerial issues surrounding RAS in Africa are explored in the following based on a systematic literature review.

The following electronic academic databases were consulted in October 2022 for relevant articles on the subject matter: Google Scholar, Scopus, PubMed, ISI Web of Science, ResearchGate and ScienceDirect, similar to the studies by Houessou et al. [23] and Guo et al. [24]. The following keywords and descriptors were used as search criteria, namely: recirculating aquaculture system, RAS, performance, profitability, challenges, opportunities,

urban farming, food security, poverty alleviation and Africa. This exercise also includes assessing the reference lists of the identified articles to ensure that relevant studies were included. The result from the database search with all keywords produced 104 citations, of which 84 were from Google Scholar and a total of 20 citations were from all other aforementioned databases. Once the literature search was complete, these 104 titles were screened for applicable papers using specific inclusion and exclusion criteria. The inclusion criteria include the continental focus, i.e., Africa, concrete estimates of RAS performance in terms of quantity, e.g., kg/m$^3$ or g/fish, profitability in monetary terms and case studies. The exclusion criteria range from non-social-science and engineering studies to those studies conducted outside Africa. See Figure 1 for the selection process steps for relevant articles. The identification process of references, screening and eligibility criteria methodology followed Guo et al. [24].

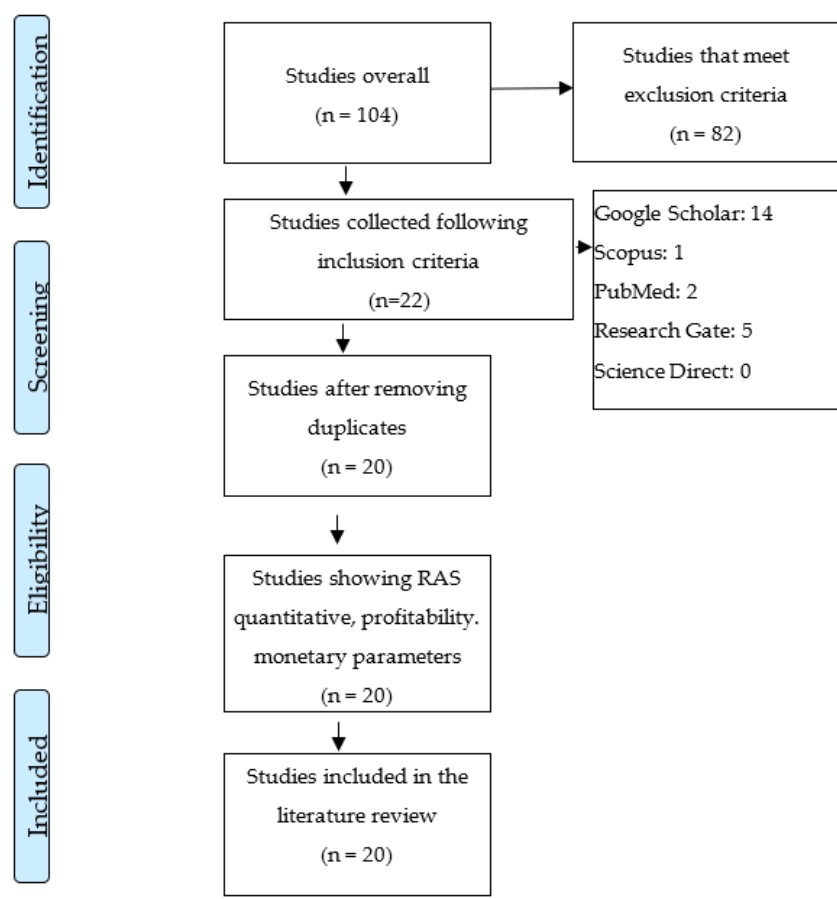

**Figure 1.** Flow diagram of literature review. *Source: Authors.*

The systematic literature review reveals that RAS is not widespread across African countries (except for South Africa and Kenya). This may explain why documentation of production processes and data are sparse [25–27]. This impression is further evidenced by the rather small number of 20 studies from the systemic literature review. Some of the African countries that have implemented RAS include Kenya, Namibia, Nigeria and South Africa, mostly for Tilapia and to some extent African Catfish [28]. The pilot project of the use of solar-powered RAS in Tilapia hatchery in Kisumu, Kenya, in 2020 resulted in the collection of 19,500 eggs from a total of 560 Nile Tilapia after 5 days valued at KSH50,000 (US$470) [29]. Munguti et al. [30] argue that RAS is efficient in reducing the grow-out period for Tilipia in Kenya, which reaches table size weight of 400–500 g within 4 to 5 months as compared to 300 g in 9 months based on conventional aquaculture. The intensive fish farms in Kenya, which use RAS for hatchery to the grow-out phase, had stocking densities of up to 125 kg/m$^3$ and an annual production of up to 200 tons in

2020 [28,30–33]. The use of RAS in Kenya is argued to require limited space, conserve water, lower feed costs by up to 50% and increase survival rates up to 25% due to water quality control compared to conventional aquaculture systems [28,34]. Wambua et al. [35] found that for RAS in Kenya, low stocking densities (2.3 kg/m$^3$ to 5.0 kg/m$^3$) resulted in longer payback periods compared to higher stocking densities between 7.0 kg/m$^3$ to 10.0 kg/m$^3$. Wambua et al. [36] argue that the performance of RAS with respect to high water quality, increased production and profitability can be improved if adequate skills on stocking biomass, power and water flow rate are made available to farmers in Kenya. In Nigeria, Akinwole and Faturoti [37] found that the high RAS stocking density of 176.6 kg/m$^3$ was possible for African Catfish due to their hardy nature, but cautioned on the prevalence of poor water quality, low growth rate and extended culture duration. Similarly, Atse et al. [38] found that for the Ivory Coast, the survival rate and biomass of African Catfish increase as the stocking density increases, but at the expense of the fish growth rate. Soliman and Yacout [39] argue that RAS is a good alternative to the conventional aquaculture systems found in Egypt, e.g., cages, earthen and concrete ponds. In comparison to conventional aquaculture in Egypt, RAS only requires 20% of the water and minimal land [39]. RAS production cycle trials in Egypt in 2010 resulted in 18.9 tons of fish with a total cost of E£ 141,368 (US$25,106) and was reported to be economically viable [40]. However, in the case of a stocking density of 250 g/fish in Tilapia RAS production in Egypt, seed or fry (small fishes) costs represented 56% of the variable costs, which is approximately seven times that of conventional production [41]. In Ghana, the use of RAS is slowly gaining traction since its introduction some eight years ago with over 150 aquapreneurs across the country using some form of a recirculating tank culture system [42]. According to Amponsah and Guilherme [42], RAS units in Ghana have the capacity of 400 Tilapia fingerlings or 1000 Catfish fingerlings (approx. at 10 g) for a table size of 300 g and 1 kg, respectively, in a span of 4–6 months. The business case study for a large-scale hatchery and grow-out with a capacity for 50 million fingerlings in Ghana in 2015 found that an annual net profit of US$1.1 million is achievable [43]. In Mmadinare, Bostwana, a large-scale RAS system was built to produce more than 500,000 fingerlings but is currently not functioning at capacity [44]. The analysis of specific farms using RAS in South Africa by Oyeleke [45] suggests that the gross profit ranges between R33,334 (US$1822) and R840,000 (US$ 45,900) for a production cycle.

To summarize, the major advantages of RAS for African countries comprise the high stocking density, often more than 100 fish per m$^3$, thus requiring considerably less water and land per kilogram of fish produced. Furthermore, operation in a controlled environment reduces losses through predators and theft [46]. However, the financial viability of small-scale RAS business models needs to be verified in the absence of subsidies [34].

According to Kaleem and Sabi [47], the major challenges for aquaculture development in Africa, which also restricts the adoption and implementation of RAS, are an inadequate supply and high cost of inputs, poor management practices, high capital and operational costs and lack of appropriate innovations. In southern Africa, inappropriate production systems, low profitability and a lack inputs were identified as the key factors for not adopting RAS for fish production [44]. According to Fornshell and Hinshaw [11], constant and stable electricity is essential for operating and managing RAS, irrespective of location. Thus, electricity is one of the major challenges foreseen for the implementation of RAS in Nigeria and throughout Africa [16]. Bodiola et al. [20] also identified power failure as a one of the major setbacks for RAS development across the globe. The lack of adequate RAS input supply chains is another major hinderrance in Nigeria [16]. According to Lutz [48], the use of non-standardized inputs can make quality control more difficult, reducing yields and thus profits. Thus, micro- and small-scale RAS tend not to thrive in (peri-)urban areas in Africa if input suppliers are not in close proximity, lack access to quality products and are unreliable.

*2.2. RAS Technology—A Technical Overview*

Compared to flow-through aquaculture, RAS is a more complex method of fish farming. RAS can be divided into several smaller sections or unit (treatment) processes that work as stand-alone unit or that are linked through a process stream. The basic concept of RAS is to have a solution (i.e., technology) and management in place for the envisaged scaled-up fish production that is effective in the sense of improving the supply of fish protein as well as being profitable within a specific region of the world. The process of a typical RAS (as illustrated in Figure 2) ensures that the water flows from a fish tank and through units that remove solids (settleable, suspended, fine and dissolved), turns the ammonia to nitrate and adds oxygen before the cleansed water is returned back to the fish tank [49]. RAS also requires a monitoring and control system to be in place to avert fish mortality due to poor water quality, diseases and other related risks.

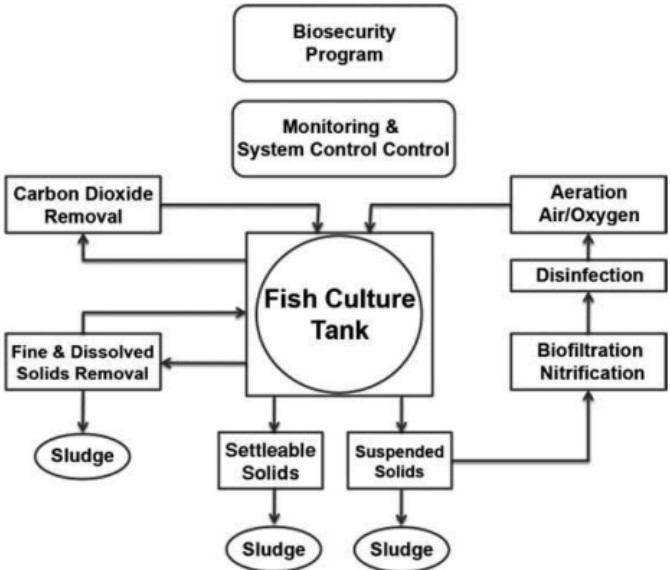

**Figure 2.** Unit processes used on a recirculating system. *Source: Ebeling and Timmons* [49] (p. 248).

The SANFU II project was implemented on an area of 13.4 m$^2$ with a partial open greenhouse in Lagos, Nigeria (see Figure 3). Relevant data such as water quality, fish weight, system management, labor input, investment and variable costs (e.g., costs of fish feed) were recorded. Market price information was collected from fish farmers and sellers in and around Lagos as well as through secondary sources. Other relevant data were collected using local knowledge if and when available. The duration of the SANFU II project data collection for this study was from March to June 2022, which corresponds to one production cycle.

The SANFU II project is based on a simplified design and fabrication of RAS consisting of a fish and sump tank, a solid lifting outlet (SLO), a redial flow settler and mechanical and biological filtration that are all linked together as a process stream (see Figure 4). The SLO pushes settleable and suspended solids into the redial flow settler, affixed with a stilling well that ensures settleable solids go to the bottom of the redial flow settler. This redial flow settler is an 80 L barrel. This is different from the Cornell dual-drain system described by Ebeling and Timmons [49] (p. 250), which uses the culture tank itself as a swirl separator ('tea-cup' effect) and removes most of the settleable solids through the center drainage. However, the results are similar in that only minimal water (10 to 25%) is used to displace settleable solids such as leftover feed and fish excrete [49]. Suspended solids are moved to the mechanical filtration system that uses a granular media filter consisting of granite rocks smaller than 1 cm in size as well as fishing nets, which intercept the solids and hinder onward flow. The use of the biological filtration system aids the nitrification process where bacterial activities convert the ammonia to nitrite and then nitrate based on the surface area

available for bacterial growth. The media used for surface areas in the biological filtration were the cap of waste plastic bottles (PEP: plastic engineered products) that were cut into four parts to act as floating beads. These floating beads further trap solids that were not intercepted in the mechanical filtration. As stated by Ebeling and Timmons [49] in the case of high levels of stocking density (>45 kg/m$^3$), similar to that of the SANFU II prototype (148 fishes is equivalent to 74 kg/m$^3$), an aeration system is needed to provide adequate levels of oxygen. The requirements for monitoring rise with increasing stock density. To this end, the regular measurements of pH and ammonia were conducted through external labor hired to oversee the biosecurity, monitoring and control procedure of the SANFU II RAS prototype. The SANFU II RAS system is equipped with a 3000 L/h water pump and 600 L/h aeration system having a power consumption of 10 Watts per hour (see Figure 5).

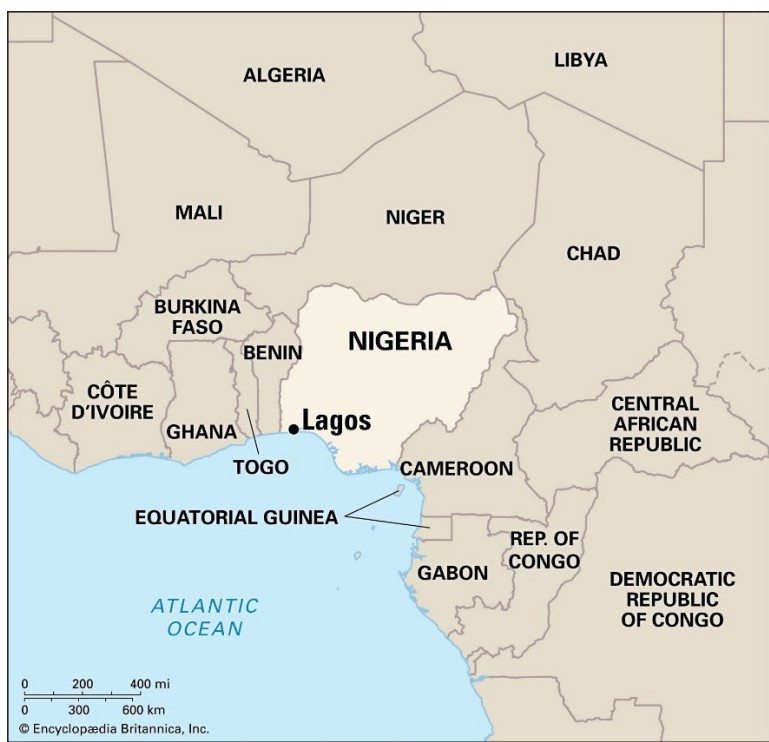

**Figure 3.** Map of Nigeria and location of Lagos State. *Source: Britannica* [50] *(p. 1).*

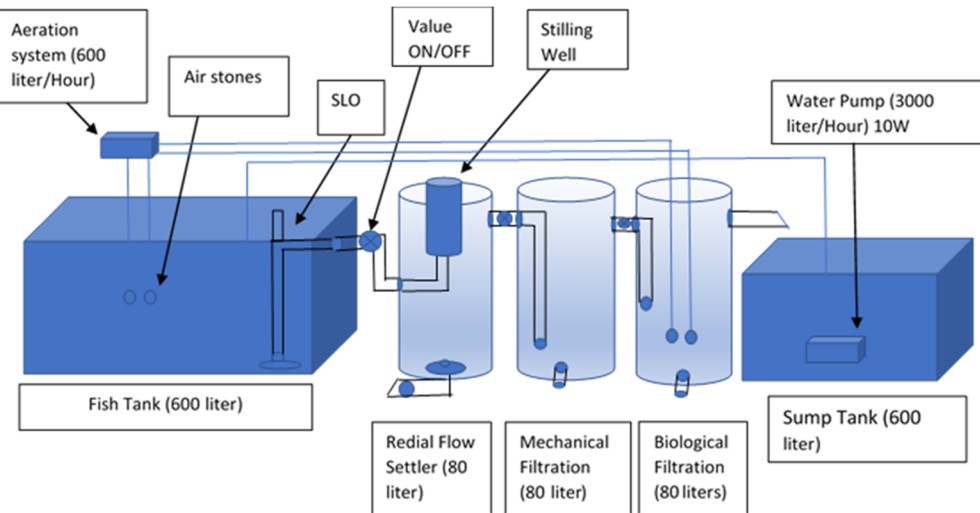

**Figure 4.** SANFU II micro- and small-scale RAS unit process design. *Notes: SLO = solid lifting outlet. Source: Authors.*

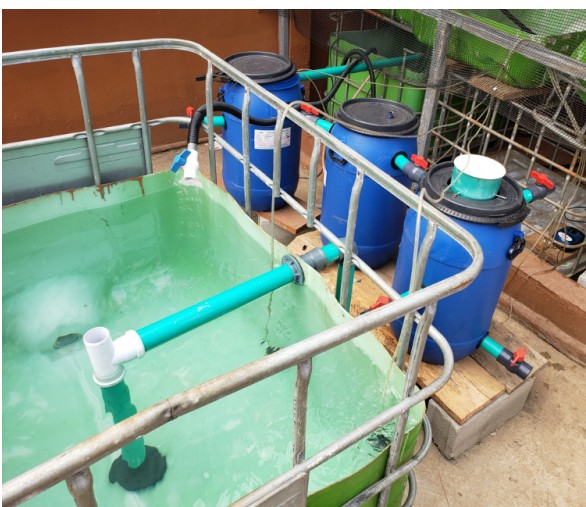

**Figure 5.** SANFU II RAS prototype implementation. *Source: Authors.*

*2.3. Efficiency and Business Management Indicators of RAS*

As earlier mentioned, the technical and financial viability of RAS depends on a number of factors ranging from water quality based on filtration, stocking density, monitoring and management of cash flow. Thus, the quality of water will be estimated based on the mass balance, i.e., the volume of solids after filtration, which is also influenced by the stocking density. The mass balance approach makes it possible to track the amount and sustainability characteristics of circular and/or bio-based contents in the parts of the whole of the supply and value chain and attribute it based on verifiable records from the management and monitoring. The cash flow analysis provides business viability estimates.

*2.4. Mass Balance*

It is important to assess the technical viability of the RAS in developing countries before looking at the cost and benefit implications. Maintaining appropriate and good water quality is essential for the successful management and operations of RAS. The quality of water can be estimated through the mass balance as well as the stocking density calculation [49]. The mass balance is denoted as:

$$Q \times C_{in} + P_{solid} = Q \times C_{out} \tag{1}$$

where $C_{in}$ and $C_{out}$ are concentrations of a vector of variables such as solids in and out of the fish tank (kg/m$^3$), $Q$ is recirculated water (liters per day) and $P_{solid}$ is the production rate of total suspended solids (TSS) (mg per liter).

$P_{solid}$ can be estimated using the mathematical formula: $0.25 \times$ kg feed fed (dry matter basis (the value of the dry matter basis ranges between 0.20 and 0.40.)).

Solving for $C_{out}$ in kg/L will provide water quality concentration for the filtration device, i.e., leftover particles in a given filter device are estimated using the mass balance analysis and are denoted as:

$$C_{out} = C_{in} + \frac{T}{100}(C_{best} - C_{in}) \tag{2}$$

where $\frac{T}{100}$ is the treatment ($T$) efficiency (%) of the filter and $C_{best}$ is the optimal result obtainable by the filtration (e.g., no suspended solids).

*2.5. Stocking Density*

When designing a RAS system, it is important to estimate the number of fishes that can be adequately and safely raised in the anticipated unit volume of the fish tanks. This should be done with the aim of moving the fish to the market once the table size weight,

i.e., usually around or above 500 g, is achieved. This pre-defined fish weight of the table size and the number of fishes in the tank are important for determining the feeding rate. Ebeling and Timmons [49] argue that the number of fishes that will be stocked for each unit volume (*density*) is based on the species and size of fish at the grow-out stage. Similarly to the approach of Ebeling and Timmons [49], this study uses the fish body length (*Length*), when the table size weight is achieved, to estimate the number of fishes that can be raised per unit volume of fish tank, denoted as:

$$D_{Density} = \frac{Length}{C_{Density}} \tag{3}$$

where $D_{Density}$ is the total weight of fish that can be stocked (or harvested) per cubic meter measured in kg/m$^3$, *Length* is the length of fish in cm and $C_{Density}$ is a default value that is dependent on the species of fish (we use a $C_{Density}$ value for the African Catfish of $0.3^4$, which is similar to that of trout species). It is, however, important to note that permissible fish stocking densities not only depend on the fish size and technical characteristics of the facility but also on operational and management skills [49].

## 2.6. System Monitoring and Management

To have a sustainable fish farming development through RAS, it is important to have an adequate management as well as a monitoring system for diseases and pathogens in place [21]. Even the small-scale RAS prototype of SANFU II requires a substantial level of monitoring and correctional measure, especially at a higher stocking density, to guarantee adequate fish wellbeing and yields. The monitoring and management tasks include filtration system cleansing and residual removal, disease control as well as controlled water replenishment. One full-time staff member, a facility manager, with on-site training in aquaculture management, was saddled with this responsibility including data recording, working six hours a day, seven days a week.

## 2.7. Cash Flow Analysis

Cash flow analysis is used to estimate the movement of funds in and out of the company account within a given timeframe. This is based on the current cash generated from operations as well as the cost incurred (fixed and variable costs) due to the running of the facility.

$$CF = R - \left( C_{fixed} + C_{variable} \right) \tag{4}$$

where $CF$ is the cash flow, $R$ is the revenue, and $C_{fixed}$ and $C_{variable}$ are fixed and variable costs, respectively. While the up-front costs of the RAS components are not relevant for the cash flow analysis, this study provided a rough estimate of the up-front costs to (peri-)urban farming entrepreneurs. The units of all the values are in currency—Naira (₦) and US Dollars (US$).

## 3. Results

### 3.1. Mass Balance

The SANFU II RAS started out with 148 African Catfish fingerlings. After 84 days into the four-month growth cycle, 22 African Catfish had achieved a marketable table size of ≥500 g. Given that the fish stock had reached its table size weight, sales activities started. Figure 6 shows the feeding pattern and decrease in stock density due to sales over time. The minimum and maximum daily feed quantity was 12 g and 1.8 kg, respectively. The average amount of daily feed was approximately 0.78 kg, with an average cost of ₦585 (US$1). The efficiency of each of the filtration systems was observed to be 98%, based on the volume of residuals of settleable solids observed in the fish tank. If the aforementioned values are computed for the concentrations of solids out, $C_{out}$, of the filtration system Equation (2), it will result in a value of 0.015 kg/L.

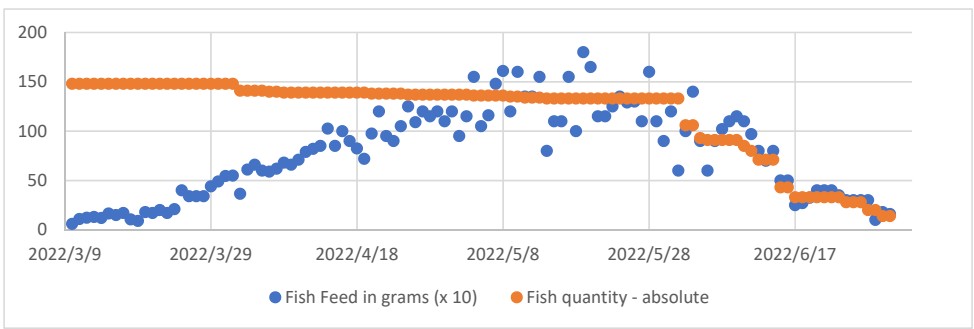

**Figure 6.** Quantity of fish and amount of fish feed in the SANFU II RAS in one cycle of 4 months. *Source: Authors*.

### 3.2. Stocking Density

The anticipated length of the African Catfish at the end of the four-month growth cycle is 40 cm. The $C_{Density}$ of 0.34 (default value used for a certain species of fish) was used for the African Catfish to estimate the stock density Equation (3) level of 118 kg/m³. Considering that the fish tank in use is 0.6 m³, this implies a stocking density, $D_{Density}$, of 71 kg/m³. Ebeling and Timmons [49] recommend to start with half of the estimated stocking density as a precautionary measure, which would correspond to 35.5 kg/m³ given the fish tank measures 600 L. This implies that 71 African Catfish with a table size of ≥500 g should be farmed in the tank. Therefore, the SANFU II project, with its initial 148 African Catfish stocking density, is twice the precautionary recommended capacity based on Ebeling and Timmons [49].

The adequate aquaculture management put in place was able to prevent a high level of mortality. The mortality rate of the SANFU II RAS was 11% and below the 12% for the Catfish mortality rate often observed in aquaculture (see UGA [51]). This is not to say there was no consequence for stocking at a relatively high density as cannibalism was observedsince only 131 Catfishes survived. This resulted in the separation and sorting of the fishes after two and half months. African Catfish above 400 g were transferred to a flow-through system and immediately sold to consumers, while those below 400 g were kept in the RAS. Figure 7 below shows the stocking density of the African Catfish in the micro- and small-scale RAS towards the end of the four-month cycle.

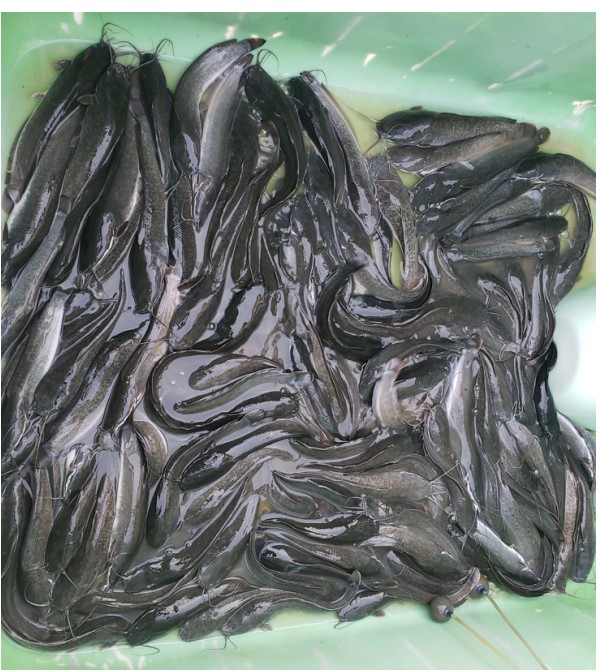

**Figure 7.** African Catfish yield of the SANFU II RAS in one production cycle of 4 months. *Source: Authors*.

### 3.3. System Monitoring and Management

The higher the high stocking density of an RAS, the higher the demands on its monitoring and management. The twice-as-high stocking density as compared to the recommended capacity based on Ebeling and Timmons [49] in the SANFU II RAS was compensated with more intense monitoring and management. This entailed cleaning the immense settleable solids and other residues from the filtration system as well as observing fish health on a daily basis. A number of events such as system clogging, water treatment and fish health concerns led to the complete removal and replenishment of the water in the fish tank over the four-month growth cycle. Figure 8 illustrates the trend in water replenishment due to filtration cleaning and the aforementioned occurrences. According to Figure 8, the 100% removal and replenishment of water took place less than 10 times in 114 days or one cycle, alongside the 10 percent water displacement attributed to filtration cleaning and effluent removal (also see Ebeling and Timmons [49]). Compared to conventional flow-through systems in which water is completely replenished every two days, depending on the stocking density, the micro- and small-scale RAS prototype conserves water. Due to the presence of adequate monitoring and management as well as the expertise of the full-time staff member (see Section 3.3) in aquaculture management, the SANFU II RAS prototype experienced a lower-than-average mortality rate. Fish mortality was predominantly due to bacterial infection. For instance, Amponsah and Guilherme [42] argue that bacterial infections account for the majority of mortalities in aquaculture. As a way of minimizing bacterial and fungal infection, a sea salt treatment and antibiotics were applied to the system once a sick fish was identified and separated. Occasionally, the sea salt treatment was also applied as a precautionary measure.

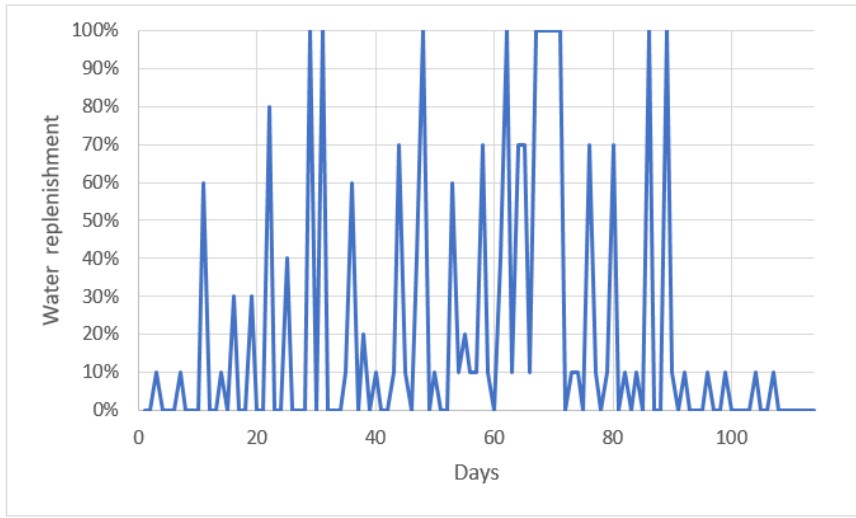

**Figure 8.** The filtration cleaning and water replenishment in the SANFU II RAS in one cycle. *Notes: Filtration cleaning always took place when the water replenishment was above 5 percent. Source: Authors.*

### 3.4. Cash Flow of the Small-Scale RAS

The RAS was designed, fabricated and implemented earlier based on donor seed money. It has been running since 2019 and was upgraded in 2022. The up-front costs of the SANFU II RAS prototype, which consist of a 600 L fish tank system (plus sorting and sump tanks), fingerlings, a 2.5 KVA solar system, 10W water pump and aeration device, were estimated at ₦700,000 (US$1200). The cost of the 2.5 KVA solar system (incl. batteries) in Lagos, Nigeria, ranges between ₦300,000 (US$517) and ₦850,000 (US$1465). Thus, the total cost of a complete solar-powered micro- and small-scale RAS would range between ₦1,000,000 (US$1724) and ₦2,000,000 (US$3500) in Nigeria. This cost range is very similar to the average annual income of between US$1046 and US$4095 per capita (in constant 2020 US$) in Nigeria, a lower–middle income country [52]. This comparison highlights the investment challenge to vulnerable groups and poor urbanites (see Benjamin et al. [16]).

Alternatively, solar energy could be sourced from a local green energy provider such as MTN Solar Electricity. The SANFU II project deploys a mixed energy approach in powering the RAS by combining off-(solar) and on-grid electricity to account for electricity needed at night. The average daily running time of the 10W water pump and aeration device was 20 h. The respective price per kilowatt-hour of solar and on-grid electricity in Lagos was ₦58 (US$0.10) and ₦45 (US$0.08), respectively [53]. The expected monthly expenditure on electricity (solar and on-grid) required to operate the RAS is estimated at ₦1233 (US$2), while the salary payable was ₦20,000 ($34). This results in a monthly fixed cost of ₦21,233 (US$36). The monthly variable cost, comprising fish feed (foreign and locally sourced), medication and other operational expenses, is estimated at ₦15,500, (US$27) bringing the total monthly fixed and variable cost of the SANFU II RAS prototype to ₦36,733 (US$63), but this can be reduced to ₦16,733 (US$29) without external labor. A more detailed overview of the total monthly fixed and variable costs is presented in Table 1. Fish feed and external labor make up 37 percent and 54 percent, respectively, of the total cost. The market price of one African Catfish at table size in Nigeria in 2022 was between ₦1000 (US$1.7) and ₦1200 (US$2.1). At the end of the four month production cycle, 115 fishes weighing above 500 g were sold, resulting in revenues of ₦130,140 (US$224). The remaining 16 fishes were used for their own consumption. This implies a monthly revenue of ₦32,535 (US$56). If the average monthly expenses are compared with the revenues, the SANFU II RAS prototype still entails a deficit of ₦4198 (US$7). In the short run, a profit will only be achievable if the paid full-time staff is substituted with family labor. The monthly cost per fish >500 g sold would thus come up to ₦111 (US$0.19) while revenue was ₦283 (US$0.49). We assume that within the family, there is a high level of unemployment given that the current unemployment rate of Nigeria is 33% [54]. These family members could pursue RAS as the alternative to forego productive capacity, losses in national income and social exclusion [55]. In this case, the monthly profit would amount to ₦19,770 (US$34) or ₦172 (US$0.30) per fish sold. It is important to note that the SANFU II micro- and small-scale RAS prototype was implemented on 13.4 m$^2$ area of land and proves that the implementation of RAS in a (peri-)urban setting in Africa requires minimal land.

**Table 1.** Overview of monthly fixed and variable cost of simplified RAS in Lagos, Nigeria.

| Fixed Cost | | | |
|---|---|---|---|
| **Description** | **Unit** | **Amount (₦)** | **Amount (US$)** |
| **Utilities** | | | |
| Solar system | KWH | 696 | 1.2 |
| On-grids Electricity | KWH | 537 | 0.9 |
| **Salary** | | | |
| Facility manager | 1 | 20,000 | 34.0 |
| **Total fixed costs** | | **21,233** | **36.1** |
| Variable Cost | | | |
| Fish feed | kg | 13,500 | 23.2 |
| Treatment of disease (antibiotics and sea salt) | kg | 1000 | 1.7 |
| Miscellaneous (Repair, replacement etc.) | 1 | 1000 | 1.7 |
| Total variable costs | | **15,500** | **26.6** |
| Total costs | | **36,733** | **62.7** |

Notes: KWH = kilowatt hour. Exchange rate applied here was US$1 = ₦580, which was the average rate observed during the field work. 600 L tank was stocked with 148 African Catfish. Source: Authors.

*3.5. Challenges of Sustainable RAS Nigeria*

The SANFU II project piloted in Lagos, Nigeria, experienced, on average, 11 h of on-grid power outages per day. The use of an alternative energy source, namely, a 2.5 KVA solar system, made the aforementioned average pump running time of 20 h possible. Figure 9 below illustrates the difference in the number of hours of available on-grid electricity and the running times of water pumps (see green arrows) supplemented by renewable energy. According to Aich et al. [21], such use of new energy sources will be vital in overcoming future challenges and attaining a sustainable blue economy.

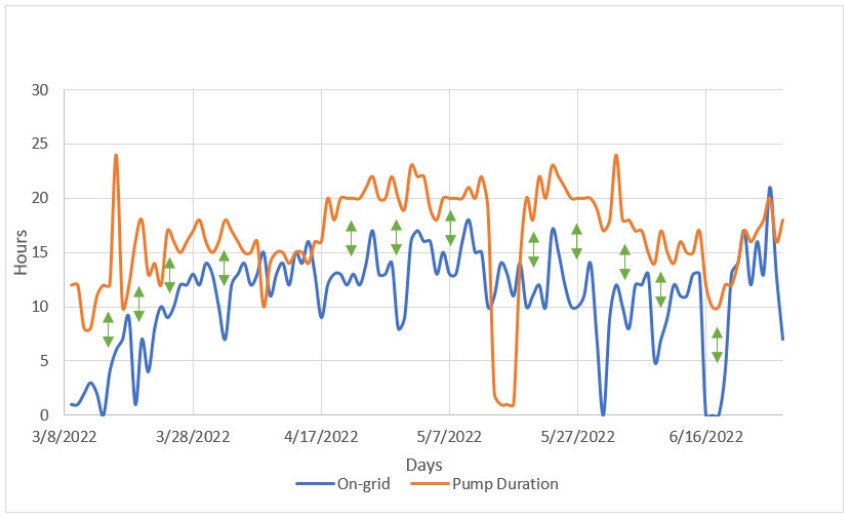

**Figure 9.** On-grid electricity supply and running time of water pumps in hours per day. Green arrows signify the energy gap covered by renewable energy due to national grid outages. *Source: Authors*.

**Input market.** There is an immense input market gap (for instance, with regard to affordable and nutritious fish feed and equipment) for RAS if the technology and practices should be adopted as part of (peri-)urban farming in Nigeria and across Africa. The situation of SANFU II project would have been worse but for the availability of some virtual marketing platforms as well as local plumbing and hardware stores. However, the substitutes and alternative equipment at plumbing and hardware stores do not necessarily conform to aquaculture standards.

## 4. Discussion

Complex system designs of small-scale RAS is one of the factors limiting the adoption and implementation of RAS in developing countries and contributing to food security as the COVID-19 pandemic, conflicts as well as climate extremes exacerbate global food insecurity and poverty for urbanites [19,20,56–63]. The shortage of skilled personnel, energy, inputs and the high-upfront cost have also made the feasibility and profitability of the system questionable, despite its ability to produce more fish per unit area compared to conventional fish farming [21]. This study sheds light on the opportunities and challenges of small-scale RAS in sub-Saharan Africa by investigating a fish production cycle and cost–benefit characteristics of a system implemented under the SANFU II project in Lagos, Nigeria, from March to June 2022 on a 13.4 m$^2$ space. The technical design of a stable RAS system should have a filtration system able to provide adequate water quality with little or no solid residue in ensuring fish health and survival [49]. The estimated water quality concentration, which is expressed in the leftover particles in a given filter device in kg/L ($C_{out}$) for the micro- and small-scale RAS of the SANFU II project, was 0.015 kg/L. This modest value attests to the efficiency of the filtration units used for fish wastewater recycling under the SANFU II project, as the value is similar to that of [48]. This system design has minimized the complete removal and replenishment of water in the RAS to less than 10 times during the production cycle compared to over 60 times in a conventional flow-through system

for the same stocking density and duration. Thus, this study contributes to the body of literature on sustainable land management and highlights the contribution of small-scale RAS to resource, e.g., water and land, conservation, as well as the reduction of effluents released into the environment.

The design of the SANFU II prototype was able to match the stocking density for African Catfish in conventional flow-through system of 71 kg/m$^3$. Similarly, Dai et al. [64] found that stocking African Catfish at a density between 35 kg/m$^3$ and 65 kg/m$^3$ provides high welfare standards, with higher stocking density hindering certain welfare indicators, such as hematological and biochemical indices. However, van de Nieuwegiessen et al. [65] argue that African Catfish can adapt to higher stocking densities of between 100 kg/m$^3$ and 300 kg/m$^3$ in intensive recirculating systems. Hengsawat et al. [66] also found that the high stocking density results in increased catfish harvests and, ultimately, higher profits. The grow out weight of the African Catfish at this stocking density at the end of the four-month period was above 500 g, with a length above 40 cm. Benjamin et al. [4,16] also found that after four months, a length of 40 cm as well as a weight of over 500 g was achievable in African Catfish. Brummet [67] argue that for aquaculture to develop in Africa and provide diverse benefits to society, a business approach that focuses on small- and medium-scale enterprises must be adopted. Thus, the SANFU II small-scale RAS model will enable practitioners, especially vulnerable groups in (peri-)urban areas, to rear fish for subsistence consumption as well as for revenue generation through sales within a short period.

Access to appropriate aquaculture monitoring and management skills is vital for RAS success. Monitoring helps to identify fish diseases and to engage in counter measures, thus reducing losses and costs. The SANFU II project trained and retained a young adult from the host community as a facility manager responsible for water quality and fish health management. The facility manager undertakes filtration maintenance as well as data collection, including the recording of fish growth, pH and ammonia of the RAS. The filtration maintenance was observed to be more frequent as fish growth progressed.

Amponsah and Guilherme [42] argue that fish farming using RAS requires a high initial investment and reliable electricity but is easy to construct on limited space, e.g., in backyard gardens or courtyards. In terms of scaling the project to reach more households, an important question to ask concerns the high initial investment, which can be mitigated through government and private sector funding, e.g., grants, concession loans, guarantee credits, etc. As mentioned earlier, the costs of small-scale RAS implementation is beyond the scope of this study. However, it is important to analyze whether small-scale RAS can sustain itself as an agribusiness once implemented. The cash flow/profitability analysis conducted for the SANFU II project follows this line of reasoning in exploring the cost–benefit of RAS in a (peri-)urban farming context. The small-scale RAS under consideration in this study can become financially viable if households utilize family labor and if the family labor has the appropriate aquaculture management skills. By replacing an external facility manager with family labor, the monthly operating expenses, estimated at ₦20,000 (US$34) of the micro-and small-scale RAS, could be reduced to ₦6000 (US$10). This will also enhance the acquisition of new skills and learning on the job. The use of family labor in (peri-)urban farming, specifically RAS, may not only improve FNS due to improved subsistence consumption, but also make it a profitable and thus viable venture. When family labor replaces external labor and aquaculture management skills are accessible, a monthly revenue of ₦32,535 (US$56) and net cash flow (profit before taxes) of ₦9802 (US$17) could be realized. While still below the daily poverty ceiling of US$1.90 a day, it actually contributes to average family income. These numbers bode well for efforts to improve FNS in (peri-)urban sub-Saharan Africa.

The challenges of small-scale RAS implementation in Nigeria, apart from the aforementioned high investment cost, are related to instable electricity and the lack of adequate inputs. A minimum of 14–16 h of electricity is required to raise hardy fish such as African Catfish or Tilapia. The electricity outages in Lagos, Nigeria, often lasting for hours, are problematic for pumping and recycling fish wastewater and providing oxygen through

aeration systems. This decreases water quality and adversely affects fish survival rate. Furthermore, the lack of inputs, specifically hardware, as well as the high variable cost of fish feed is a major concern for small-scale RAS adoption and implementation in Nigeria. Conventional plumping hardware, often used in aquaculture operations in Nigeria, reduces the efficiency of RAS.

*Policy Implication*

For RAS to be financially viable, acquiring the proper aquaculture management skills is important. Despite the importance of these skills as well as the growing recognition of the role that aquaculture could play in fostering FNS, reducing environmental pressure due to ASF production, aquaculture management skills remain mostly unavailable when consulting the list of extension services provided by extension workers in sub-Saharan Africa [68,69]. Getting aquaculture management skills on the radar of public extension workers is therefore essential, especially for households who might be unable to afford private extension services.

Other areas where policymakers could improve the viability of RAS concern how to reduce upfront and energy costs. Governments should not only prioritize (peri-)urban farming in short- and long-term agriculture and development agendas and city planning, but work on optimizing the already existing energy infrastructure to improve its efficiency. Investing in renewable energy sources (see Jacal et al. [70]) to ease power availability concerns and energy costs of operating RAS systems represents a cost-effective approach. Governments in sub-Saharan Africa could invest in renewable energy to increase the availability of green electricity and make them accessible to the public. Incentives include subsidy provisions, reducing regulatory constraints to encourage private investments, developing a stable regulatory framework that reduces environmental pollution, e.g., introducing a carbon tax can help to mobilize massive private sector investments in renewable energy sources [71].

Finally, technology has an important role to play in further enhancing the availability of RAS equipment and inputs, for instance, online platforms that could improve access to aquaculture inputs and RAS technology. Another useful area is in linking RAS technology developers to potential users to facilitate dialogue between individuals who respectively design and use the technology to create a feedback network. This represents a cost-effective way for users to gain needed expertise to operate the technology, and feedback from users to developers could provide insights on how to further optimize the functionality of RAS technology. An example is the linkage of RAS practitioners and stakeholders to digital innovation hubs (DIHs) such as the SmartAgriHubs.eu or DigitalAgriHubs.eu. This will provide them with access to digital decision support and risk analysis tools as well as access to investors. Public and private policymakers should create supportive frameworks to encourage the development of these platforms. This could be done by providing funding to develop and sustain the platforms, leveraging already existing networks, e.g., extension agencies, to promote and market the platforms to a wider audience and building strong public–private sector partnerships to attract more investors for RAS systems.

## 5. Conclusions

The majority of fish produced in African (peri-)urban areas is from flow-through aquaculture. Flow-through aquaculture is unsustainable in city region food systems because it requires substantial land and water resources and pressures the environment through effluents. Sustainable (i.e., circular and resilient) and equitable city region food systems with strong production and market connections are a critical foundation for FNS, thriving communities and businesses [5,72,73]. This can be achieved by transforming linear production and midstream components of the city region food system into circular ones [22]. Circularity in a food system context implies reducing the amount of waste generated and changing diets towards more diverse and resource-efficient food patterns. RAS are circular and suitable for the unique context of African cities because they do not require great

access to land, water or wealth. RAS has the potential to produce more fish per unit area compared to conventional fish farming. Yet, RAS has not witnessed a broad adoption and implementation in (peri-)urban farming in developing countries. This is attributed to high up-front costs, complex system designs, unstable electricity, limited aquaculture managerial skills, etc.

This study assessed the technical and financial viability of a simplified small-scale RAS prototype stocked with 148 African Catfish and implemented under the SANFU II project from March to June 2022 in Lagos, Nigeria [49]. The low fish mortality rate is attributed to the efficient filtration system as well as adequate monitoring and management of the system. Assuming that the monitoring and management is taken over by a qualified family member, a unit profit of ₦172 (US$0.30) can be achieved. Alternatively, the produced fish could be consumed by the family, thus reducing the purchasing costs for fish protein and contributing to improved FNS.

These results imply that small-scale RAS are technically and financially viable if labor costs are moderate, e.g., through employing paid family labor with proper aquaculture monitoring and management skills. Furthermore, access to adequate equipment and inputs as well as electricity for the recirculating system is crucial. (Peri-)urban innovation actors will only adopt RAS if they are efficient, have low capital and operating costs, and are profitable.

**Author Contributions:** E.O.B.: conceptualization, methodology, formal analysis, writing-original draft, writing-review & draft, funding acquisition; O.O.: formal analysis, writing-review & editing; G.R.B.: validation, writing-review & editing, supervision. All authors have read and agreed to the published version of the manuscript.

**Funding:** This research received no external funding.

**Institutional Review Board Statement:** Not applicable.

**Informed Consent Statement:** Not applicable.

**Data Availability Statement:** Not applicable.

**Acknowledgments:** We would like to thank the management team (www.aglobedc.org, accessed on 3 September 2022) and staff of Aglobe Development Center, Lagos, Nigeria, especially Dare Balogun and Sulaimon Babalola. We also extend our thanks to Tolulope Olatunbosun of RFisheries Nigeria Limited and Diaspora expert program of the Deutsche Gesellschaft für Internationale Zusammenarbeit GmbH (GIZ).

**Conflicts of Interest:** The authors declare no conflict of interest.

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
