# Peer review of "Feasibility Study of a Small-Scale Recirculating Aquaculture System for Sustainable (Peri-)Urban Farming in Sub-Saharan Africa: A Nigerian Perspective"

_land, doi:10.3390/land11112063_

Round 1
Reviewer 1 Report (New Reviewer)
Dear Authors,
I found your article very interesting and useful expecially for developing areas. It is very technical in some parts but well written, the methodology is clear as the results well explained. The content is good, however, I found something in the structure that maybe should be changed, and some part has just to move to other places, but it is correct to me. Please find below my suggestions and I hope they can help your work.
· I would remove the acronyms from the title
· Please correct the numbering of the bibliography, the first number that appears must be 1 and not 28. The bibliography should be not in alphabetical order but in order of appearance
· line 51: it is not from [15] but from De Bon et al (2010) [15]
· Sub-paragraph 2.1 should be paragraph 2 “Literature review” and “materials and methods” become paragraph 3 (and so on). Please, evaluate that even subparagraph 2.2 should move in paragraph 2, but it is your choice
· In materials and methods paragraph a map of Nigeria with the area affected by the project would help
· In general, I think there are too many subparagraphs, you should simplify the number of them and try to harmonize the speech better in order to be less schematic
· In the footnote: put single line spacing
· line 124: give space between 2.1 and “the”
· figure 2: the name of the author in the source should be indicated
· Formula 1: maybe it is better to replace X with *
· Line 304: 4 is superscript
· line 379: put author name before [40] (also on line 429, 576)
· 3.4 has the same title as 2.7
· Part of the sub-paragraph 4.1 related to policy implications should be moved from paragraph 4 and placed in the conclusions where you can put more emphasis on the political implications of the project, the rest of 4.1 should be summarized a little and you can make a single paragraph for discussions.
Author Response
Paper title:
“Technology-Business-Management of Recirculating Aquaculture System (RAS) for Sustainable Urban Farming in Sub-Saharan Africa: A Nigerian feasibility study”
We thank the reviewers for their comments and suggestions. They have motivated us to rethink and clearly articulate the contribution of our paper. The revisions to our paper have been done accordingly.
The revisions are highlighted in the manuscript. In the following, we explain how we have addressed the concerns raised and incorporated the suggestions into this revised version. We cite where we effected these changes in the updated manuscript.
Remarks of reviewer 1
Comments to Author
Overview and general recommendation
I found your article very interesting and useful expecially for developing areas. It is very technical in some parts but well written, the methodology is clear as the results well explained. The content is good, however, I found something in the structure that maybe should be changed, and some part has just to move to other places, but it is correct to me. Please find below my suggestions and I hope they can help your work.
- I would remove the acronyms from the title
We have now removed the acronym – RAS. Now reads –
“Technology-Business-Management of Recirculating Aquaculture System for Sustainable Urban Farming in Sub-Saharan Africa: A Nigerian feasibility study”
- Please correct the numbering of the bibliography, the first number that appears must be 1 and not 28. The bibliography should be not in alphabetical order but in order of appearance
This has been revised.
- line 51: it is not from [15] but from De Bon et al (2010) [15]
This has been revised.
- Sub-paragraph 2.1 should be paragraph 2 “Literature review” and “materials and methods” become paragraph 3 (and so on). Please, evaluate that even subparagraph 2.2 should move in paragraph 2, but it is your choice
This was the recommendation given by the previous reviewers for the structure the paper.
- In materials and methods paragraph a map of Nigeria with the area affected by the project would help
A map of Nigeria and the state the project is located is now inserted in the manuscript. See lines 237 -240
- In general, I think there are too many subparagraphs, you should simplify the number of them and try to harmonize the speech better in order to be less schematic
This were the recommendations given by the previous reviewers for the structure of the paper
- In the footnote: put single line spacing
All footnotes are already single line spaced
- line 124: give space between 2.1 and “the”
This has been revised
- figure 2: the name of the author in the source should be indicated
This has been revised
- Formula 1: maybe it is better to replace X with *
This has been revised
- Line 304: 4 is superscript
This has been revised
- line 379: put author name before [40] (also on line 429, 576)
This has been revised
- 3.4 has the same title as 2.7
This has been revise to reflect that the former is the methodology description while the latter is the result
- Part of the sub-paragraph 4.1 related to policy implications should be moved from paragraph 4 and placed in the conclusions where you can put more emphasis on the political implications of the project, the rest of 4.1 should be summarized a little and you can make a single paragraph for discussions.
It was recommended by the previous reviewers that policy implication should be embedded in the discussion section.
Reviewer 2 Report (New Reviewer)
In the case that there is an increasing demand for animal sourced food, this paper considered the impact of small-scale Recirculating Aquaculture Systems (RAS) on improve food improvement and nutritional security, livelihoods as well as reduce environmental degradation in (peri-)urban areas. This study also assessed the RAS prototype of the Sustainable Aquaponics for Nutritional and Food Security in Urban Sub-Saharan Africa (SANFU) II project. This article is of great practical significance.
However, there are also some problems of this study.
1. Figure 1: Flow Diagram of Literature Review looks a little simple. Perhaps the author should add some explanations to the treatment.
2. My main consider is that the body of this paper seems to be a specification of an item. The author should emphasize its application and value. And the relevance of this topic to land.
Some minor mistakes:
i. page 4 line 139-140, “Google Scholar, Scopus, PubMed, ISI Web 139 of Science, ResearchGate, and ScienceDirect similar to the studies by [70, 71]. ” This article is of great practical significance. There are many similar sentences, which reduce the readability of the article
Author Response
Paper title:
“Technology-Business-Management of Recirculating Aquaculture System (RAS) for Sustainable Urban Farming in Sub-Saharan Africa: A Nigerian feasibility study”
We thank the reviewers for their comments and suggestions. They have motivated us to rethink and clearly articulate the contribution of our paper. The revisions to our paper have been done accordingly.
The revisions are highlighted in the manuscript. In the following, we explain how we have addressed the concerns raised and incorporated the suggestions into this revised version. We cite where we effected these changes in the updated manuscript.
Remarks of reviewer 2
Comments to Author
Overview and general recommendation
In the case that there is an increasing demand for animal sourced food, this paper considered the impact of small-scale Recirculating Aquaculture Systems (RAS) on improve food improvement and nutritional security, livelihoods as well as reduce environmental degradation in (peri-)urban areas. This study also assessed the RAS prototype of the Sustainable Aquaponics for Nutritional and Food Security in Urban Sub-Saharan Africa (SANFU) II project. This article is of great practical significance.
However, there are also some problems of this study.
- Figure 1: Flow Diagram of Literature Review looks a little simple. Perhaps the author should add some explanations to the treatment.
These are the first steps in a systemic review methodology – also see Houessou et al. [22] and Guo et al. [23] – We do not understand exactly what the reviewer means by explanations to the treatment
- My main consider is that the body of this paper seems to be a specification of an item. The author should emphasize its application and value. And the relevance of this topic to land.
The application of this innovation is the design of an efficient filtration system and proper management that allow urban farmers to raise fishes above the precautionary recommendations (see Ebeling and Timmons argument). The recommendations from this study are also in line with the capacity of small-scale conventional flow-through system (line 496 – 500).The idea is to show that RAS can exceed the precautionary recommended fish density with a good filtration and management skill so as not to result in a deficit. There are several value added of the innovation, specifically environmental – water and land conservation and livelihood (see lines 378 – 472). The relevant of the topic for land is highlighted throughout the manuscript (see lines, 75 – 80, 174 -176, 444 – 447, 493-495)
Some minor mistakes:
- page 4 line 139-140, “Google Scholar, Scopus, PubMed, ISI Web 139 of Science, ResearchGate, and ScienceDirect similar to the studies by [70, 71]. ” This article is of great practical significance. There are many similar sentences, which reduce the readability of the article
This aspect has been revised throughout the manuscript
Reviewer 3 Report (New Reviewer)
This manuscript investigates the advantages of using small-scale Recirculating Aquaculture Systems (RAS) for aquaculture compared to flow-through aquaculture and its potential application in sub-Saharan Africa. Taking SANFU II as an example, the authors have conducted long experiments and observations to demonstrate its technical and financial advantages in terms of water quality based on filtration, stocking density, monitoring, and management to cash flow, which provide a technical approach to achieve the sustainable development of fisheries. The manuscript is well structured, with reasonable methods and results of practical value. However, the central concern of this study “the key opportunities and challenges of implementing small-scale RAS in (peri-)urban farming contexts from a technical, business and managerial perspective” is not in the scope of this journal. Although it explores an adaptive way to aquaculture and appears to be relevant to the scope of the special issue, it has nothing to do with the objectives of the section on "Socioeconomic and Political Aspects of Land" or the aims of the journal, but focuses more on the equipment SANFU II itself. Therefore, regret to say that I do not think this paper is suitable for this journal, although it is well written and I believe the authors will be able to find a suitable journal for publication.
Author Response
Paper title:
“Technology-Business-Management of Recirculating Aquaculture System (RAS) for Sustainable Urban Farming in Sub-Saharan Africa: A Nigerian feasibility study”
We thank the reviewers for their comments and suggestions. They have motivated us to rethink and clearly articulate the contribution of our paper. The revisions to our paper have been done accordingly.
The revisions are highlighted in the manuscript. In the following, we explain how we have addressed the concerns raised and incorporated the suggestions into this revised version. We cite where we effected these changes in the updated manuscript.
Remarks of reviewer 3
Comments to Author
Overview and general recommendation
This manuscript investigates the advantages of using small-scale Recirculating Aquaculture Systems (RAS) for aquaculture compared to flow-through aquaculture and its potential application in sub-Saharan Africa. Taking SANFU II as an example, the authors have conducted long experiments and observations to demonstrate its technical and financial advantages in terms of water quality based on filtration, stocking density, monitoring, and management to cash flow, which provide a technical approach to achieve the sustainable development of fisheries. The manuscript is well structured, with reasonable methods and results of practical value. However, the central concern of this study “the key opportunities and challenges of implementing small-scale RAS in (peri-)urban farming contexts from a technical, business and managerial perspective” is not in the scope of this journal. Although it explores an adaptive way to aquaculture and appears to be relevant to the scope of the special issue, it has nothing to do with the objectives of the section on "Socioeconomic and Political Aspects of Land" or the aims of the journal, but focuses more on the equipment SANFU II itself. Therefore, regret to say that I do not think this paper is suitable for this journal, although it is well written and I believe the authors will be able to find a suitable journal for publication.
Response to Reviewer 3
We thank the reviewers for their comments and suggestions. The central concern of this study “the key opportunities and challenges of implementing small-scale RAS in (peri-)urban farming contexts from a technical, business and managerial perspective” does fit the scope of this journal. This is because we have been able to show that efficient and resource conserving small-scale RAS system the can be implemented in (peri-)urban areas of developing countries despite land constraints. This is even possible on land areas as small as 13m2 (lines 458 – 460). This is important as it provides food and nutrition security to vulnerable groups during external shocks such as the COVID pandemic or international conflicts (lines 64 – 72, 486 – 490). The focuses is not on the equipment of SANFU II but a blueprint for developing countries on small-scale RAS system implementation based on international standards (lines 256 – 350).
Round 2
Reviewer 2 Report (New Reviewer)
Thanks to the author for revising the paper.
On this version, more relevant content added to the literature. Some language errors were also corrected. All these changes have improved the quality of the article. But some sentences are still incomplete, for example, line 81-82, line262-263, page numbers of the literature are emphasized. Other than that, no more problems.
Author Response
We thank the reviewers for their comments and suggestions.
Remarks of reviewer 2
Comments to Author
Overview and general recommendation
On this version, more relevant content added to the literature. Some language errors were also corrected. All these changes have improved the quality of the article. But some sentences are still incomplete, for example, line 81-82, line262-263, page numbers of the literature are emphasized. Other than that, no more problems.·
Line 81-82 is a direct citation in quotation marks. This sentence is therefore complete, as it cannot be altered. The line 262-263 has been revised and now reads: This is different from the Cornell dual-drain system described by Ebeling and Timmons [48] (p. 250) which uses the culture tank itself as a swirl separator (‘tea-cup’ effect) and removes most of the settleable solids through the center drainage.
Reviewer 3 Report (New Reviewer)
Thanks for the author's reply. As you mentioned, RAS and its potential applications in Sub-Saharan Africa provide a way for smallholder farming to adapt to future crises. The manuscript is very well written especially its "Methods" and "Results" sections. It is recommended that the authors add more discussion on the value and application of this research to land economics and sustainable land management in the Discussion part as a separate subsection (similar to 4.1 Policy Implication).
Author Response
We thank the reviewers for their comments and suggestions.
Remarks of reviewer 3
Comments to Author
Overview and general recommendation
Thanks for the author's reply. As you mentioned, RAS and its potential applications in Sub-Saharan Africa provide a way for smallholder farming to adapt to future crises. The manuscript is very well written especially its "Methods" and "Results" sections. It is recommended that the authors add more discussion on the value and application of this research to land economics and sustainable land management in the Discussion part as a separate subsection (similar to 4.1 Policy Implication).
The first part of the discussion section deals with the value and application of this research to land economics and sustainable land management. The sentence, lines 506 to 510, has been revised to emphasize the value and application of this research to land economics and sustainable land management.
This manuscript is a resubmission of an earlier submission. The following is a list of the peer review reports and author responses from that submission.
Round 1
Reviewer 1 Report
Title: Technology-Business-Management of Recirculating Aquaculture System (RAS) for Sustainable Urban Farming in Sub-Saharan Africa: A Review of Challenges and Opportunities
Recommendation
Reject, with maybe possibility to rework the whole paper and resubmit as a new manuscript.
Comments to Author
Overview and general recommendation
This paper touches upon an important subject of providing fish protein in rapidly growing African cities to ensure food security and nutrition. This subject of providing fresh food (vegetables, fish, etc.) in general has recently gained more recognition with the COVID19 pandemic and related measures that tested the resilience of local and global food systems against shocks. Therefore, locally developed interventions such as Recirculating Aquaculture System (RAS) in urban areas are helpful in softening the blow to food security in these difficult times.
The paper has the merit to work on that issue. However, it would have contributed to knowledge if the paper had clearly justified the geographical focus, conducted a literature review on challenges and opportunities, and clarified what to improve to promote RAS adoption.
Major comments
1. The article seems to be more of a feasibility study rather than a research paper. The research question is not clear.
2. The main concern I have about this manuscript is the promotion of a technology that is not profitable. In fact, the technology seems to be profitable only if the worker managing it works for free. It is unfair to promote injustice while addressing a development issue. Even if, there is a need to develop small scale technologies for urban farming in SSA, it is not acceptable to engage a family member for free before making profits because as indicated by the paper, the goal is to make profit while contributing to FNS.
3. Another concern is on the geographic focus of the paper. Is it Nigeria or SSA? The introduction nicely focused on SSA, but the following sections clearly showed that the research was conducted only in Nigeria with no research activity like a literature review that clearly sought for evidence from other parts of the African continent.
4. Another concern is the singularity of the methodology that only analysed the technical and financial viability of the technology. However, the introduction announced the need to review evidence from the whole SSA, so I was expecting a proper conduct of a literature review to collect and analyse evidence on the performance, challenges and opportunities of RAS utilization. I roughly found a small subsection on two challenges in the results with no indication of where there were sought. In addition, the discussion revealed some challenges and opportunities that were never presented in the results section. Reading the following material may help you define a more detailed methodology to complement the current results of the paper: Houessou, M. D., van de Louw, M., & Sonneveld, B. G. (2020). What Constraints the Expansion of Urban Agriculture in Benin? Sustainability, 12(14), 5774.
Minor comments:
Title. The title of the paper is confusing and misleading. I was really expecting a literature review in the methodology. Depending on the revisions made to the paper, you must consider revising the title to adjust it to the new directions given to the storyline.
Lines 38 and 53 and elsewhere. Give also examples of other countries. Since the paper focused on Sub-Saharan (SSA) countries, giving the only examples of Nigeria to support your arguments seems unfair to other SSA countries. Also, giving examples of various countries from various parts of SSA easily justify how the issues discussed here are regional.
Line 104. “……., are limited”. Please, do a proofreading of the text before final acceptance.
Lines 105-114. Although, it is nice to give a sense of the main findings in the introduction, there is already too many details here about the technical and financial findings that I would prefer to see in the results. Also, I would prefer that the last paragraph introduces the outline of the various sections of the paper.
Line 173. “…Ranging from…..to cash flow”. Does it mean that there factors not yet cited in this sentence? It is important to indicate clearly what factors are considered in the analysis here.
Section 2.5. System monitoring. What is the recommended labour per area? It is is important to clarify the baseline used to indicate whether the One full time staff here is sufficient or not.
2.6. Cash-flow analysis. Why the up-front costs of the RAS components are not relevant for cash flow analysis? How are R, Cfixed and Cvariable are estimated and what are their components.
Line 254. What happened with the 128 Catfish used in the experiment?
Cash flow analysis. You made the calculations with 115 fiches which is still beyond the 71 recommended. Do you then recommend exceeding the recommendations, why? Why do you recommend to use family member since the entrepreneur can employ a worker and allocate his or her time to expand the business. Concretely, by exceeding the recommended fish density, there is still a deficit in using RAS, so, why would RAS be adopted if it is profitable? Also, why there is only an estimation of costs in the table 1. It is important to see every calculations in the table with the two scenarios suggested (with and without a worker). On another angle, the world is fighting to promote decent jobs for poor people, so, there is need to clarify why would you recommend a system that uses a full-time employer without paying for his time. There are many questions that would always discourage the use of such an innovation. How can it be improved to cover all costs and make profit.
Section 3.1. why this is the first section? Please, also put above indicators calculated under a subsection. The most important concern in this section is how the arguments are built? It is never indicated in the methodology that you performed a systematic review or a literature survey to inform us on the challenges? So, I was surprised to see a literature consultation. Furthermore, how can we be sure that there are only two challenges impeding RAS adoption? It is crucial to indicate in the methods how you conducted this literature review? Moreover, the title and introduction introduced that the paper would also tease out the opportunities: where are they? Or did I miss it somewhere? Please, clarify all of these.
Lines 354-368. This first paragraph of the discussion repeats the rationale of the study instead of answering the main question posed in the introduction: Why RAS has not witnessed broad adoption in urban farming in sub-Saharan Africa ? insights on the challenges and opportunities
Lines 396-402. In addition to electricity, high investment was also indicated as a constraint but it was never presented in the results. Next, the possibility to construct on limited space was announced as an opportunity but nothing is developed on opportunities in the results while it was a announced as a main focus of the paper.
Lines 410-421. There are some efforts to demonstrate that RAS is profitable if a family worker is employed without payment and that this would increase family income. The question is where would the family find upfront investments to install the RAS system if it is not profitable in the end. It is difficult in these conditions to expect a high adoption. What can be changed to RAS to make it accessible and profitable.
Reviewer 2 Report
This manuscript reported the RAS in Africa from business and technology management viewpoint. Some issues should be clarified to improve quality of this manuscript.
1 Why did this study only investigate 4 months (March of 2022 to June of 2022)? Authors mentioned that they reviewed the RAS so that more periods should be included.
2 Fig 6: Are these data obtained by authors or just collect from the references? If from references, please add the citation.
3 Fig 7: Are these data obtained by authors or just collect from the references? If from references, please add the citation.
4 Only "3.1. Challenges of sustainable RAS in urban sub-Sahara Africa." was shown. Where are section 3.2, 3.3...? The authors are suggested to rearrange sections in part III.
5 Conclusion is too long. Please shorten this part and keep the key points.